# Dynamic Design of a Quad-Stable Piezoelectric Energy Harvester via Bifurcation Theory

**DOI:** 10.3390/s22218453

**Published:** 2022-11-03

**Authors:** Qichang Zhang, Yucheng Yan, Jianxin Han, Shuying Hao, Wei Wang

**Affiliations:** 1Tianjin Key Laboratory of Nonlinear Dynamics and Control, School of Mechanical Engineering, Tianjin University, Tianjin 300350, China; 2Tianjin Key Laboratory of High Speed Cutting and Precision Machining, Tianjin University of Technology and Education, Tianjin 300222, China; 3Tianjin Key Laboratory for Advanced Mechatronic System Design and Intelligent Control, School of Mechanical Engineering, Tianjin University of Technology, Tianjin 300384, China

**Keywords:** energy harvester, multi-stability, bifurcation modes, geometric nonlinearity

## Abstract

The parameter tuning of a multi-stable energy harvester is crucial to enhancing harvesting efficiency. In this paper, the bifurcation theory is applied to quantitatively reveal the effects of structural parameters on the statics and dynamics of a quad-stable energy harvester (QEH). Firstly, a novel QEH system utilizing the geometric nonlinearity of springs is proposed. Static bifurcation analysis is carried out to design quad-stable working conditions. To investigate the cross-well and high-energy vibration, the complex dynamic frequency (CDF) method, suitable for both weakly and strongly nonlinear dynamic problems, is then applied to deduce the primary response solution. By using the unfolding analysis in singularity theory, four steady-state properties and dozens of primary resonance modes are demonstrated. Based on the transition set, the effective bandwidth for energy harvesting can be customized to adapt well to various vibration environments by parametric adjustment. Finally, the experimental tests verify that the output power can reach up to 1 mW. The proposed QEH and its mechanics optimization can guide energy supply for next-generation wireless systems and low-power sensors under magnetic forbidding environments.

## 1. Introduction

A vibrational energy harvester is a kind of micro-energy generation device that captures vibrational energy from human or mechanical motion and supplies electrical energy for the micro-power embedded electronics and wireless sensors [1,2] so as to reduce the energy supply cost.

In its early stages, the piezoelectric energy harvester (PEH) was based on the resonant mechanism [3], which is not suitable for a broadband frequency environment and time-dependent frequency excitation. In order to overcome the narrow frequency band, various broadband mechanisms have been achieved, including array structured spread spectrum techniques [4,5], active or passive intermittent tuning techniques [6] and pre-load tuning techniques [7]. Since active control increases power consumption and array layout occupies significant space, nonlinear techniques have been proposed and proven to be superior [8,9]. Wang et al. [10] proposed a compact ultralow-frequency and broadband T-shaped PEH that achieved an output power of 605 μW under the external excitation of 11 Hz and 0.5 g. Yang et al. [11] combined the flextensional transducer and nonlinear elastic beam and proposed a compressive-mode harvester which generated a power of 19 mW at 21 Hz under harmonic excitation with a peak acceleration of 0.5 g. Fan et al. [12] explored a mono-stable energy harvester with an attractive magnetic force for the substantial enhancement of energy from excitations, which exhibits a wider bandwidth and a significantly larger peak voltage than the linear PEH. Wang et al. [13] designed a piezoelectric wind energy harvester with Y-shaped attachments on the bluff body, which confirmed the transition from vortex-induced vibration to galloping. Considering that both vortex-induced vibrations and the galloping effect have a desirable impact on high amplitudes, Ambrożkiewicz et al. [14] proposed a PEH whose tip-mass is a mixed design of the bluff body with different shapes of cross-section. This device can extract energy from its environment at lower air velocity values.

Since the introduction of multi-stability, the advantages of multiple potential wells have been fully developed and utilized. Zhou et al. [15] used nonlinear magnetic force and magnetic stoppers to adapt to the low-frequency (<10 Hz) weak excitation, and the available average power was 29.5 μW under a constant harmonic excitation amplitude of 0.5 g at 8.5 Hz. Stanton et al. [16] proposed a bi-stable energy harvester comprised of a piezoelectric cantilever beam and permanent magnets. In order to further improve environmental adaptability, a weaker energy barrier is required. Wang et al. [17] constructed a novel tri-stable galloping-based PEH aimed at flow-induced vibrations. Additionally, various vibration modes were explored, such as intra-well, inter-well and chaotic vibration. Zhou et al. [18] analyzed the harvesting performance influence mechanism of the asymmetry of potential wells and then changed the equilibrium positions to obtain different dynamic characteristics of tri-stable energy harvesters, which can efficiently harvest energy under various excitation conditions. Wang et al. [19,20] theoretically and experimentally studied a piezoelectric vibration quad-stable and quin-stable (which has five stable static equilibria in its system) energy harvester induced by the combined nonlinearity of cantilever–surface contact and magneto-elasticity. However, in some applications of wireless sensor networks, the magnetic field could strongly interact with the micro-sensors [21,22,23]. Additionally, a magnet-based oscillator could be easily interfered with or even locked-up by the ferromagnetic material in the host structure. To solve this issue, the geometric nonlinear energy harvesters have been proposed. Younesian and Alam [24] demonstrated that a nonlinear restoring force can be generated by using oblique springs, which represent a typical geometrical nonlinearity. Yang and Cao [25] proposed a new type of electromagnetic piezoelectric hybrid bi-stable and tri-stable energy harvesters based on the SD vibrator with a snap-through mechanism [26,27], where the geometric nonlinearity is adopted to design multi-stable structures and optimize the dynamic performance.

The aforementioned literatures on multi-stable systems showed that system parameters have a considerable effect on the stable-steady number of energy harvesters. By varying structural parameters, the system can present different multi-stabilities. Additionally, the adjustment of structural parameters could also affect the effective bandwidth. Fan et al. [28] found that the bandwidth of the PEH with stoppers can be shifted left by changing the gap between the tip mass-magnet and the external magnets. Mei et al. [29] theoretically analyzed the qualitative influence of the nonlinear magnetic parameters on the output power of the proposed harvester. However, such analyses are not sufficiently detailed. Using the singularity theory, Zhang et al. [30,31,32] acquired the transition sets of the finger spacing-comb separation plane and the DC–AC voltage plane of the MEMS gyroscope, and divided the corresponding parameter planes into six regions, which give the available parameter regions that meet the working requirements of the gyroscope. Hou et al. [33] carried out the two-state variable singularity method and derived several different bifurcation modes of the nonlinear parameters and the damping ratio.

Currently, the quantitative influence of structural parameters on the dynamic response of the PEH remains uninvestigated. This paper proposes and investigates a geometric nonlinear energy harvester with a quad-stable state which has no magnetic field generation. To improve the efficiency of the energy harvesting, the influence of the structural parameters on the amplitude–frequency response of the system is emphatically studied using the bifurcation theory, and these results are then used to define the effective bandwidth of the harvester. The above analysis has important applications for the design and optimization of dynamic systems. Through the according adjustment of structural parameters, the energy harvester can adapt to vibration sources in different frequency bands. The validity of the theory model is verified by the experimental study.

## 2. Modeling Analysis and Numerical Calculation

### 2.1. Structural Description

Firstly, we designed a quad-stable energy harvester (QEH), as shown in Figure 1. The energy harvester consists of a beryllium bronze beam with a piezo-patch and three hinge-connected springs which are compressed or stretching. When the beam structure is subjected to external vibration *p*(*t*), with the bending of the beam, the piezo-patch derives strain and generates electrical energy. The additional constraints at the free end of the beam provided by the three springs (with equal spring stiffness *k* and uncompressed length *L*) make the system gain geometric nonlinearity, thus promoting work efficiency for the harvester.

Based on the literature [16], the governing dynamic equation of the QEH can be expressed as:(1)$$\ddot{x}(t)+2\xi \omega_{0}\dot{x}(t)+\omega_{0}^{2}x(t)-F(x)-\theta v(t)=-\Gamma \ddot{p}(t)$$
(2)$$C_{p}\dot{v}(t)+\frac{v(t)}{R_{L}}+\theta \dot{x}(t)=0$$
where ξ is the damping ratio; $\omega_{0}$ is the natural frequency of the beam; θ is the piezoelectric coupling coefficient; Γ is the external excitation coefficient and $C_{p}$ is the capacitance; *F*(*x*) is the stress at the end of the beam under the displacement *x*, which is given by [25]: (3)$$\;F=-k(x(t)-b)\left(1-\frac{L}{\sqrt{{\left(x(t)-b\right)}^{2}+h^{2}}}\right)-k(x(t)+b)\left(1-\frac{L}{\sqrt{{\left(x(t)+b\right)}^{2}+h^{2}}}\right)-kx(t)\left(1-\frac{L}{\sqrt{h_{c}{}^{2}+x{\left(t\right)}^{2}}}\right)-k_{0}x(t)$$

Due to the complexity of the control equations, it is difficult to directly derive the steady-state number of the system through the homogeneous form of Equations (1) and (2). In order to determine the relationship between the system parameters and steady-state characteristics, static bifurcation is introduced to obtain the transition set.

### 2.2. Static Equilibra

Bifurcation falls into two classes: static and dynamical, depending on the object of study. Generally, the static bifurcation is used to study the change in the number of solutions caused by the change of parameters [34], which can be meaningful to understand system static properties and multi-stability. The singularity theory established by Golubitsky and Schaeffer [35] is an effective approach to investigating the identification and classification of static bifurcation problems.

Observe that *F* is an odd function of *x*: we focus on the bifurcation that occurs at the trivial solution *x* = 0 in Equation (3). The Maclaurin expansion is applied to simplify Equation (3). By expanding as a Maclaurin series, a simple algebraic expression with *Z2*-symmetry can be obtained as follows:(4)$$g(x,\lambda ,\beta_{1},\beta_{2})=x^{7}-\lambda x+\beta_{1}x^{3}+\beta_{2}x^{5}$$
where $\left\{\beta_{1},\beta_{2},\lambda \right\}$ is transformed from the structural parameters $\left\{h,h_{c},b\right\}$ that we are most interested in, and the entire expressions are given in Equations (5)–(8). Other physical parameters in the equation have little influence on the steady-state characteristics, and can be considered constants. For simplicity, take {$\beta_{1}$,$\beta_{2}$} as the unfolding parameters, and take λ as the bifurcation parameter.
(5)$$\gamma =\left(\frac{5hkL}{16h^{8}}-\frac{\left(64b^{6}h^{2}-240b^{4}h^{4}+120b^{2}h^{6}-5h^{8}\right)kL}{8{\left(b^{2}+h^{2}\right)}^{15/2}}\right)$$
(6)$$\lambda =\left(k-\frac{kL}{h}+2k\left(1+\frac{b^{2}L}{{\left(b^{2}+h^{2}\right)}^{3/2}}-\frac{L}{\sqrt{b^{2}+h^{2}}}\right)+k_{0}\right)/\gamma$$
(7)$$\beta_{1}=\left(\frac{hkL}{2h^{4}}-\frac{\left(4b^{2}h^{2}-h^{4}\right)kL}{{\left(b^{2}+h^{2}\right)}^{7/2}}\right)/\gamma$$
(8)$$\beta_{2}=\left(-\frac{3hkL}{8h^{6}}-\frac{3\left(8b^{4}h^{2}-12b^{2}h^{4}+h^{6}\right)kL}{4{\left(b^{2}+h^{2}\right)}^{11/2}}\right)/\gamma$$

The singularity theory is then employed to analyze the effects of {β1,β2} on the equilibria of Equation (4). Different parameter regions are divided by transition sets, and these regions can be expressed as ∑(ℤ2)=B(ℤ2)∪H(ℤ2)∪DL(ℤ2), where B, H and DL are bifurcation sets, hysteresis sets and double limit point sets, respectively. Their forms are defined by Hou et al. [33]:B={β∈Rn:∃(x,λ,β), s.t.g=gx=gλ=0 }H={β∈Rn:∃(x,λ,β), s.t.g=gx=gxx=0 }DL={β∈Rn:∃(xi,λ,β) (i=1,2,…,n) ,x1≠x2, s.t.g=gx=0 }

To understand the characteristics of the singularity set ∑(ℤ2), the intersection line is obtained by the parallel section method. Figure 2 shows the transition set of the plane β1−β2. The solid, dashed and dotted lines are the hysteresis set H_0_, H_1_ and the double limit point set DL, respectively. The plane β1−β2 is divided into four persistent regions by the three lines. The bifurcation modes have the properties of topological equivalence and perturbation preservation in every independent region. There are three types of bifurcation modes in the system: supercritical pitchfork bifurcation (β1=0, β2>0), subcritical pitchfork bifurcation (β1=0, β2<0) and saddle-node bifurcation at the hysteresis set *H*_1_. Since the occurrence of bifurcation can change the coordinates and number of static equilibria, it can guide the system design and optimization to achieve superior performance. In the design of energy harvesters, a large number of steady states is the desired design goal. By analyzing the bifurcation of the parameters *h* − *x*, the different steady-state characteristics can be found in Figure 2. In order to discuss the various bifurcation modes in detail, four bifurcation diagrams are shown in Figure 3, which reflect the four independent regions of the transition set. The horizontal preassembled length *h* of the inclined spring is taken as the variable to show the coordinates *x* of the static equilibria. The solid and dashed lines are the stable and unstable equilibria in pictures, respectively. In the I diagram, the supercritical pitchfork point PFsup and the saddle node points SN1,2,3 can be observed. These bifurcation points classify the system into mono-, bi- and quad-stable states. When the unfolding parameters evolve into diagram II, the change of the bifurcation points results in the system obtaining a tri-stable region. In diagrams III and IV, the disappearance of the bifurcation point causes the steady-state number of the system to decrease sequentially. It is noteworthy that the changes in the bifurcation state necessarily lead to a critical state of the system. For example, when the unfolding parameters in the transition of I → II satisfy β1,β2∈DL, points PFsup and SN1 occur under the same conditions (consider symmetry, g=gxi=0 , x1=0, x2=xSN). Analogously, two critical conditions H_0_ and H_1_ occur in the transition of III → II transition and the transition of I → IV transition. In this case, the bifurcation point *PF* satisfies g=gx=gxx=0 , which is also the critical state for the transition between the supercritical pitchfork bifurcation PFsub and the subcritical pitchfork bifurcation PFsub. According to the design criteria of multi-stable energy harvesters, the two steady-state spaces, I and II, are parameter spaces of concern. The numerical points directly calculated from Equation (4) anastomose the bifurcation curves, which means that the application of the bifurcation theory allows for the fast, comprehensive and accurate acquisition of the target parameter space in the design of micromechanical devices.

The focus is now on the parameter spaces where the quad-stable and tri-stable states occur. By referring to the bifurcation theory to select reasonable structural parameters, the dynamical behaviors of the system with tri-stability and quad-stability are investigated. Figure 4 shows the time history diagrams and phase diagrams solved by the fourth-order Runge-Kutta method. The quad-stable behavior is evident in Figure 4a,b, so as the tri-stable behavior in Figure 4c,d. It should be noted that harmonics and chaos in the response are ignored, and different intra- or inter-well motions can be derived from different initial conditions. Through increasing the external input energy, the oscillator can escape the single-well and achieve global inter-well vibrations, which can generate higher output power. Therefore, it is an important goal in the design of energy harvesters to keep the potential energy barriers as weak as possible in order to enable the harvester to rapidly reach the energy threshold required for the inter-well motion.

## 3. Dynamic Analyze and Bifurcation Investigation

In engineering and practical applications, the selection of the frequency band is considered a significant target of the dynamic design. Taking excitation frequency as a bifurcation parameter and structural parameters as unfolding parameters can solve the design problem we are facing most directly and significantly.

The complex dynamic frequency (CDF) method proposed by Wang et al. [36] is an effective method for solving strongly nonlinear vibration problems. This method via complex normal form introduces a dynamic frequency factor to convert the differential equations into a set of algebraic equations which could deal with the effect of higher-order nonlinear terms. Zhang et al. [37] proved that the CDF method is suitable for harvesters which feature the complicated nonlinear mechanism, such as multi-stability and piecewise linearity.

### 3.1. Primary Amplitude–Frequency Curve

Considering that Equation (3) contains complex irrational terms, which are difficult to solve using the analytical method, it is therefore necessary to transform irrational terms into the approximation polynomial *u*. According to the orthogonal projection theorem, Equations (1) and (2) can be converted into:(9)$$\ddot{x}+\omega_{0}^{2}x-\kappa v=u(x,\dot{x})+f\cos(\Omega t)$$
(10)$$\;\mu v+\dot{v}+\kappa \dot{x}=0$$
where $u(x,\dot{x})=\alpha_{1}x+\alpha_{3}x^{3}+\alpha_{5}x^{5}+\alpha_{7}x^{7}+2n\dot{x}$. The displacement *x*, velocity $\dot{x}$ and voltage *v* can be expressed as:(11)$$x=\zeta +\bar{\zeta }+b,\;\dot{x}=i\omega (t)\left(\zeta -\bar{\zeta }\right),v=\sum_{i=1}^{4}\left(\Lambda_{2i-1}+i\Lambda_{2i}\right)\zeta^{2i-1}+\left(\Lambda_{2i-1}-i\Lambda_{2i}\right){\bar{\zeta }}^{2i-1}$$
where $\zeta =ae^{i\omega_{10}t+i\phi }/2$, $\zeta =ae^{i\omega_{10}t+i\phi }/2$, $\omega (t)=\omega_{10}+\sum_{k=1}^{n}\delta^{k}\omega_{1k}(t)$. Substituting Equation (11) into Equations (9) and (10) and balancing the coefficients of each term yields:(12)$$\left\{ \begin{array}{l} \cos(\phi )=\frac{1}{64F}(35\alpha_{7}a^{7}+40\alpha_{5}a^{5}+48\alpha_{3}a^{3}+64\alpha_{1}a-64a\Gamma_{1}\kappa +64a\omega_{0}{}^{2}-64a\omega_{10}^{2})\\ \sin(\phi )=\frac{1}{F}a(n\omega_{10}-\Lambda_{2}\kappa ) \end{array}\right.$$

The amplitude-frequency response curve is obtained by eliminating ϕ: (13)$${\left(35\alpha_{7}a^{7}+40\alpha_{5}a^{5}+48\alpha_{3}a^{3}+64\alpha_{1}a-64a\Lambda_{1}\theta +64a\omega_{0}^{2}-64a\omega_{10}^{2}\right)}^{2}+64{\left(an\omega_{10}-a\Lambda_{2}\theta \right)}^{2}-64F^{2}=0$$

To obtain the stability of solutions, the perturbation variables $a=a_{0}+\Delta a$, $\;\phi =\phi_{0}+\Delta \phi$ are introduced. By substituting them into Equation (12) and differentiating with respect to *t*, the first-order approach equation is given as:(14)$$\left\{ \begin{array}{l} \frac{d\Delta a}{dt}=-{\Delta a\frac{\partial \Psi (a,\omega )}{\partial a}|}_{a_{0}}+{\Delta \phi \;\Omega (a,\omega )|}_{a_{0}}\\ \frac{d\Delta \phi }{dt}=-{\Delta a\frac{\partial \Omega (a,\omega )}{\partial a}|}_{a_{0}}-{\Delta \phi \;\Psi (a,\omega )|}_{a_{0}} \end{array}\right.$$
and the expression of the eigenvalue equation is as follows:(15)$$\lambda^{2}+({\Psi (a,\omega )|}_{a_{0}}+{\frac{\partial \Psi (a,\omega )}{\partial a}|}_{a_{0}})\lambda +({{\Psi (a,\omega )|}_{a_{0}}\frac{\partial \Psi (a,\omega )}{\partial a}|}_{a_{0}}+{\Omega (a,\omega )|}_{a_{0}}{\frac{\partial \Omega (a,\omega )}{\partial a}|}_{a_{0}})$$

If the Δa and Δϕ are asymptotically stable, the domain $\left(a\;,\omega_{10}\right)$ satisfies the following inequalities:(16)$$\left\{ \begin{array}{c} {{\Psi (a,\omega )|}_{a_{0}}\frac{\partial \Psi (a,\omega )}{\partial a}|}_{a_{0}}+{\Omega (a,\omega )|}_{a_{0}}{\frac{\partial \Omega (a,\omega )}{\partial a}|}_{a_{0}}>0\\ {\Psi (a,\omega )|}_{a_{0}}+{\frac{\partial \Psi (a,\omega )}{\partial a}|}_{a_{0}}>0 \end{array}\right.$$

From the above inequalities, it is clear that the stability of solutions is independent of the external excitation amplitude *F*. Then, a set of parameters are randomly employed to demonstrate the existence of multi-solvability, but the parameter domain is not easy to intuitively determine. Figure 5 illustrates the amplitude–frequency curves with different excitation amplitudes. Equation (16) is represented by the dashed lines, indicating the unstable solutions. In this figure, the system has a maximum of seven periodic solutions, which contain four stable and three unstable solutions. 

### 3.2. Amplitude Bifurcation

Because the periodic solutions of the nonlinear vibration (determinate solution) are static in nature [34], the amplitude bifurcation remains a static bifurcation. In this paper, the excitation frequency is directly taken as the bifurcation parameter and {*α*_1_, *α*_3_, *α*_5_, *α*_7_} are taken as unfolding parameters. Instead of universal unfolding, Equation (13) is analyzed as the bifurcation equation, so that the difficulty of back substitution caused by excessive folding can be avoided.

The singularity theory consists of three aspects: the recognition, the unfolding theory and the classification problem. In this work, only the unfolding analysis of singularity theory is used to derive the bifurcation of the dynamic response. The concrete process can be found in Han et al. [38].

Taking the substitution $\lambda =\omega_{10}^{2}$, $\varepsilon =-64F^{2}$. The left side of Equation (13) can be regarded as an unfolding of the dynamic system, and the bifurcation equation is given as:(17)$$G(r,\lambda ,\varepsilon ,\alpha_{1},\alpha_{3},\alpha_{5},\alpha_{7})={\left(35\alpha_{7}r^{3}+40\alpha_{5}r^{2}+48\alpha_{3}r+64\alpha_{1}+64\lambda +\rho \right)}^{2}r-\varepsilon$$
where $\rho =-64\omega_{10}^{2}-64\Lambda_{1}\theta +64(n\omega_{10}-\Lambda_{2}\theta )$, which are physical parameters that are of no concern to us. Derive Equation (14) in the following order:(18)$$\begin{array}{c} G_{x}=2r\left(48\alpha_{3}+80r\alpha_{5}+105r^{2}\alpha_{7}\right)\left(64\lambda +\rho +64\alpha_{1}+48r\alpha_{3}+40r^{2}\alpha_{5}+35r^{3}\alpha_{7}\right)\\ +{\left(64\lambda +\rho +64\alpha_{1}+48r\alpha_{3}+40r^{2}\alpha_{5}+35r^{3}\alpha_{7}\right)}^{2} \end{array}$$
(19)$$\begin{array}{c} G_{xx}=4\left(48\alpha_{3}+80r\alpha_{5}+105r^{2}\alpha_{7}\right)\left(64\lambda +\rho +64\alpha_{1}+48r\alpha_{3}+40r^{2}\alpha_{5}+35r^{3}\alpha_{7}\right)\\ +r\left(\begin{array}{l} 2{\left(48\alpha_{3}+80r\alpha_{5}+105r^{2}\alpha_{7}\right)}^{2}+2\left(80\alpha_{5}+210r\alpha_{7}\right)\\ \left(64\lambda +\rho +64\alpha_{1}+48r\alpha_{3}+40r^{2}\alpha_{5}+35r^{3}\alpha_{7}\right) \end{array}\right) \end{array}$$
(20)$$G_{\lambda }=128r\left(64\lambda +\rho +64\alpha_{1}+48r\alpha_{3}+40r^{2}\alpha_{5}+35r^{3}\alpha_{7}\right)$$

According to the singularity theory, the transition set of the primary resonance can be calculated according to the expressions of the various non-persistent sets {B, H, DL} mentioned in Section 2.2. It is a six-dimensional hypersurface in the control space, which cannot be drawn visually. For the purpose of visualization and simplified representation, the same parallel section method is used to find the intersection line. The following is an example of $\alpha_{1}-\alpha_{3}$ plane transition set, and the same analysis can be performed for other unfolding parameters.

As shown in Figure 6, the transition set of plane $\alpha_{1}-\alpha_{3}$ contains two hysteresis sets H_1_, H_2_ and two double-limit sets DL_1_, DL_2_. Four curves divide the plane into 31 persistent sets and 47 critical states (non-persistent sets), and each area corresponds to a unique bifurcation mode. To prevent tedious presentation, Figure 7, Figure 8 and Figure 9 only show bifurcation diagrams for the partial bifurcation modes, which are representative and significant. In these figures, we marked the limit points, which would convert into hysteresis points in non-persistent set conditions. The dashed lines with the same color represent that the system exhibits a softening or hardening characteristic in a certain frequency band.

Figure 7 shows the amplitude–frequency response curves for regions G and E. The system exhibits the hardening characteristic (red segments) in these regions. Interestingly, the bifurcation patterns in Figure 7a,b are distinct because of the appearance of hysteresis set H_1_. Point a in Figure 7c is considered a critical state where $G=G_{x}=G_{xx}=0\;$ is satisfied, which corresponds to the H_1_ line between regions *G* and *E* in Figure 6 rather than a persistent region. Similarly to the Duffing Equation, for a given excitation frequency, there exist a maximum of three periodic solutions in the system. 

In addition, the amplitude–response curves for regions L3 and K3 are shown in Figure 8. In Figure 8a, the curve bends to the right and then to the left. It can be noted that both hardening characteristic and softening characteristics are observed, while the softening characteristic is completely contained by the hardening one. Figure 8b bends to the right again, giving the system a new hardening characteristic (green segment), which leads to the system having up to seven periodic solutions in this case. Similarly, there is a critical state between regions L3 and K3, which is caused by $G_{xx}=0$ at point d. This corresponds to the H_2_ line between regions L3 and K3 in Figure 6. Obviously, the combination of the parameters makes the system exhibit a vibration behavior, as shown in the e–f bandwidth, that can greatly increase the power generation for the harvester, since the system undergoes a large periodic motion crossing the four potential wells in this band. Furthermore, the stable periodic solution, as evidenced in Figure 5, indicates that the broadening of this band is effective for the optimization of the energy harvester. Thus, making the e–f band cover a wider bandwidth is a significant issue.

Equation (13) implies that increasing the strength of the excitation intensity can broaden the bandwidth of the four-well vibration. However, in this paper, the enhanced bandwidth of the inter-well motion can also be achieved at the low-intensity excitation level by tuning the parameters. Figure 8 gives the responses of regions K1–K2and K4–K5and the corresponding critical states under the same external excitation. The amplitude-frequency response curve of region K4 is plotted in Figure 9, wherein the hardening hysteresis band (band a and b) partially overlaps with the softening hysteresis band (band c–e), and the hardening hysteresis band (band d–f) is completely contained by the narrow softening hysteresis band (band c–e). By appropriately adjusting of the parameters to region K5, as shown in Figure 9b, the e–f bend can be broadened. This is because the limit points e and f are gradually shifted to the outward sides as the system transitions from region K4 to region K5. Additionally, the further shifting of points e and f makes the e–f band wider, as shown in the amplitude–frequency response curves in Figure 9c,d. In the evolution of the amplitude–frequency curve with the parameters, the critical case for line DL_2_ occurs, as shown in Figure 9e, where f and b are located at the equivalent frequency. Similar situations are observed when regions K5–K2and K2–K1are separated by the line DL_1_, corresponding to the critical case shown in Figure 9f,g, respectively. Generally, the presence of the hysteresis set H denotes the appearance (or disappearance) of new softening or hardening characteristics, and the double limit point set DL marks the broadening (or narrowing) of the softening or hardening bandwidth.

## 4. Experiment Research

### 4.1. Experimental Setup

The experiments were conducted to check the dynamic analysis and evaluate the performance of QEH. As shown in Figure 10, the prototype experiment consisted of an acceleration sensor (model: aepe, sensitivity: 10.16 mV/g, Endevco, California, United States), a displacement sensor (model: IL100, Keyence, Osaka Japan), a digital oscilloscope (model: DSOX1204G, Keysight Technologies, Beijing, China), a signal analyzer (model: 3039, Brüel & Kjær, Copenhagen, Denmark), and an excitation unit (model: APS113, SPEKTRA, Dresden, Germany). The excitation pattern is sweep frequency excitation, and the load resistance is equivalent to a resistance box (*R_L_* = 150 kΩ).

In order to undergo the maximum strain, a Micro Fiber Composite (MFC, M-2807-P2, Harbin Core Tomorrow Science & Technology Co., Ltd., Harbin, China) was attached to the root of the beam. The beam still has more space for patches, which could further improve the efficiency of the harvester.

### 4.2. Characterization

As a verification of the dynamical analysis, region K1 shown in Figure 5 was chosen as the prototype structural parameters. Experimental and theoretical results at an acceleration of 0.6 g over the frequency band 5–17 Hz are given in Figure 11a, where the combination of structural parameters for the prototype is taken in the K1 parameter space. Due to the domination of the chaos and superharmonic in the motion, the experimental curve and the primary resonance response curve do not match at 5 Hz. As the frequency increased, the system then jumped to a large inter-well motion at 6.3 Hz and began to periodically cross all potential wells, which increased the response amplitude and output voltage. The theoretical solution was confirmed by the experiment in this frequency band. The periodic motion continued until the excitation frequency increased to 11.6 Hz. At 11.6–13.4 Hz, the subharmonic amplitude was larger than the primary resonance amplitude, which led to a deviation of the experimental result from the theoretical solution of the primary resonance.

The output voltage of the prototype whose parameters were selected in the K1, L3 and G regions are shown in Figure 11b. In order to clearly display the high power band, Figure 11b only retains the data points of inter-well motions. The output voltage for the K1, I3, L3 and G regions are 28 V, 27 V, 14 V and 4 V, respectively, with average output powers of 1.02 mW, 1.01 mW, 0.62 mW and 0.17 mW, which would be sufficient to power most micro-power sensor devices. Additionally, the difference between K1 and I3 shows that the change in structural parameters enables the harvester to output the high power within different bandwidths. Furthermore, the disparity between the output power corresponding to the three parameter selection regions is enormous, which adequately demonstrates that the parameter tuning has a significant impact in terms of performance. For different vibration environments, we can adjust the corresponding parameters to make the harvester conduct the high amplitude in corresponding bandwidths, so that it could have broad adaptability to the environment.

## 5. Conclusions

In this paper, a quad-stable piezoelectric energy harvester is designed and a detailed analytical procedure is given. The bifurcation modes of a QEH system were investigated. The key findings are summarized as follows:
(1)In view of complex irrational resilience, the corresponding universal unfolding for static bifurcation analysis are determined and the parameter space of the four steady states is revealed. Bifurcation modes for each state are described in detail, including three different static bifurcation points (i.e., supercritical pitchfork bifurcation, subcritical pitchfork bifurcation, and saddle node bifurcation).(2)The parameter spaces of 31 persistent sets and 47 critical states are determined. From the amplitude–frequency response curve, we found that the presence of the hysteresis set may provide additional periodic solutions, and the double limit point set may enable the movement and expansion of a certain hardening or softening characteristic frequency band. These independent bifurcation modes bring more potential for the design of the energy harvester.(3)The experimental results show that the output power of the prototype can reach 1.02 mW at the optimum parameters. Such power is sufficient to supply most micro-power electronics and makes it possible to realize the next generation of self-powered technology.

The results obtained in this paper contribute to a better understanding of the significant effect of parameters tuning in energy harvester systems, and lay the foundations for enhancing the efficiency of energy harvesting. The derivation of the parameter transition set provides guidance for engineering designs, which is helpful to design the multi-well, high-energy vibration, and wide-bandwidth energy harvesters.

Further work will focus on an effective approach to identify the critical parameters dominating the dynamical bifurcation. Since it is difficult to visually analyze the hypersurfaces in the high-dimensional bifurcation equations, another method is to reduce the dimension through the Lyapunov-Schmidt (L-S) reduction method, and then study the bifurcation of the reduced system with the singularity theory.

## Figures and Tables

**Figure 1 sensors-22-08453-f001:**
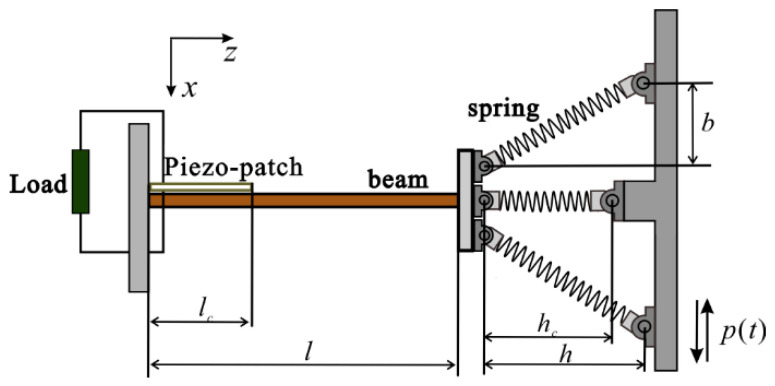
Schematic presentation of QEH.

**Figure 2 sensors-22-08453-f002:**
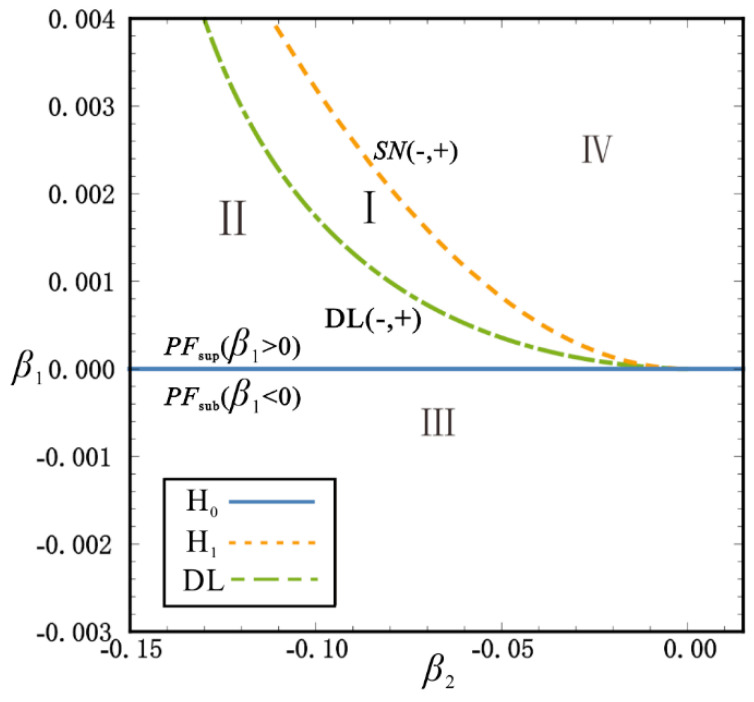
Transition set on the *β*_1_ − *β*_2_ plane.

**Figure 3 sensors-22-08453-f003:**
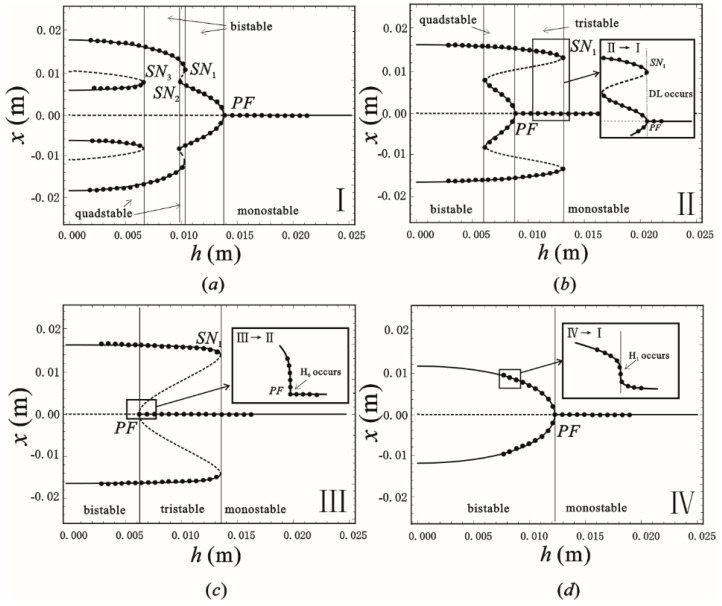
Bifurcation diagrams of the equilibrium solutions in four persistent regions, where block diagrams are critical cases of H_0_, H_1_ and DL and scattered points represent numerical results. (**a**) region I; (**b**) region II; (**c**) region III; (**d**) region IV.

**Figure 4 sensors-22-08453-f004:**
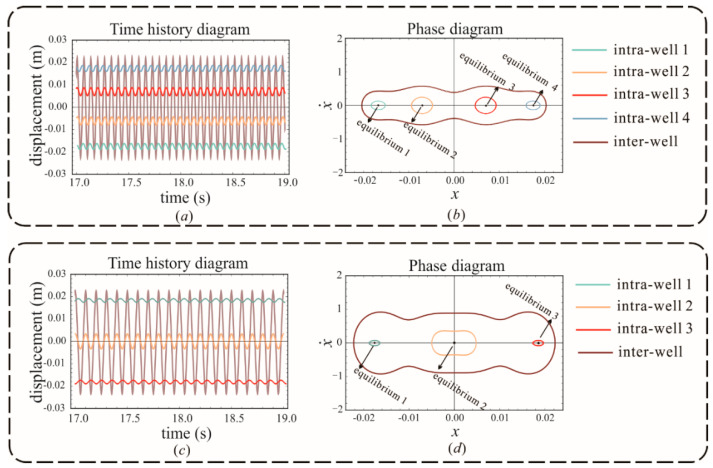
Time history and phase diagram of the (**a**,**b**) quad-stable system and (**c**,**d**) tri-stable system.

**Figure 5 sensors-22-08453-f005:**
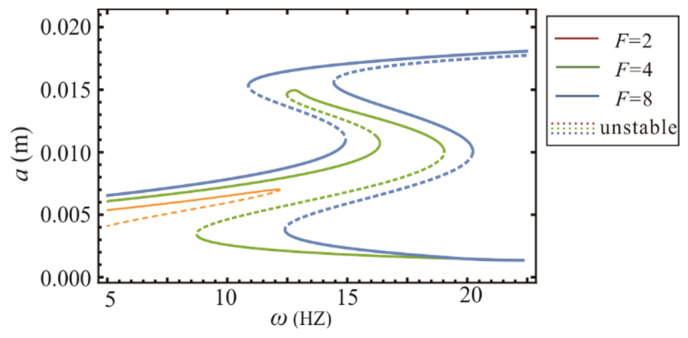
The stability of the solutions at different amplitudes *F*, and the dashed lines are the unstable solution which is introduced by Equation (16).

**Figure 6 sensors-22-08453-f006:**
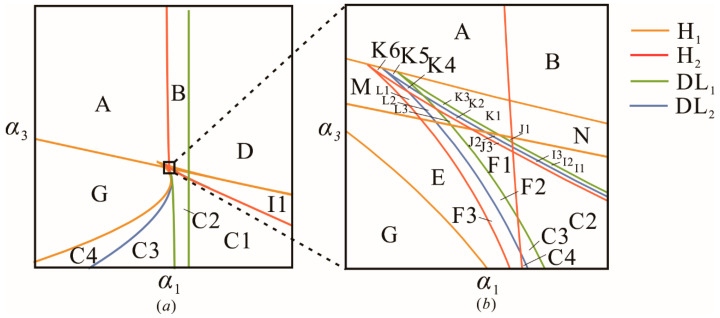
(**a**) *α*_1_ − *α*_3_ plane transition set of the system for primary resonance. (**b**) Partial enlarged view.

**Figure 7 sensors-22-08453-f007:**
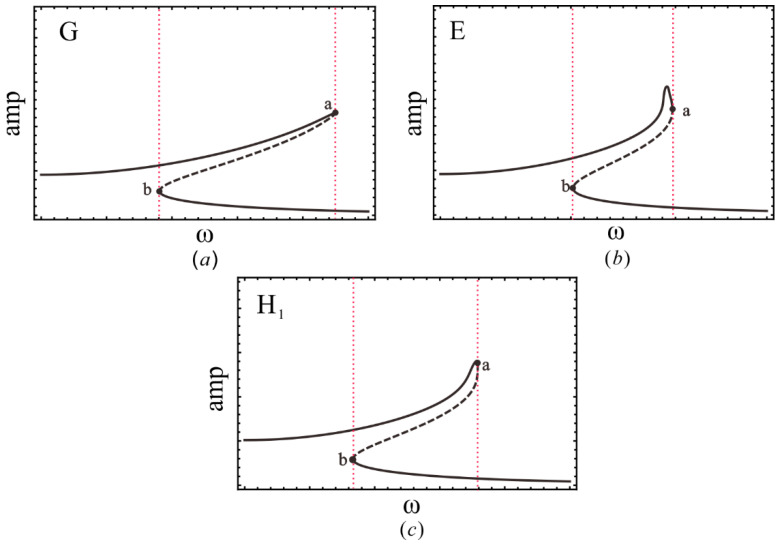
Amplitude-frequency response curve in (**a**) region *G*; (**b**) region E; and (**c**) line H_1_, where the solid points a, b are tangent points.

**Figure 8 sensors-22-08453-f008:**
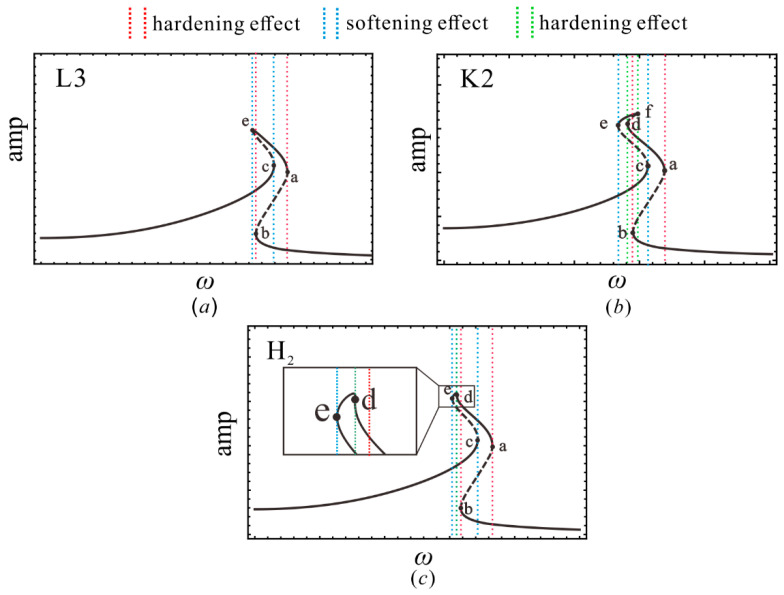
Amplitude-frequency response curve in (**a**) region L3; (**b**) region K2; and (**c**) line H_2_, where the solid points a–f are tangent points.

**Figure 9 sensors-22-08453-f009:**
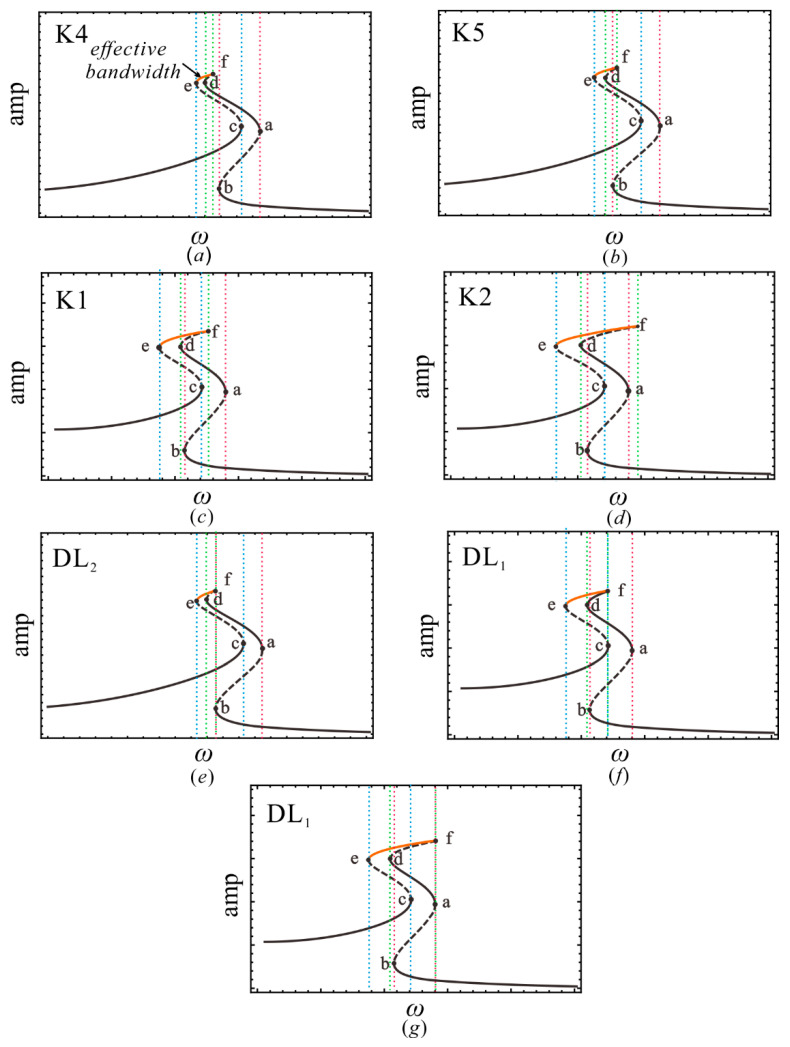
Amplitude–frequency response curve in (**a**) region K4; (**b**) region K5; (**c**) region K1; (**d**) region K2; (**e**) line DL_2_ of K4–K5; (**f**) line DL_1_ of K5–K2; (**g**) and line DL_1_ of K2–K1, where the solid points a–f are tangent points.

**Figure 10 sensors-22-08453-f010:**
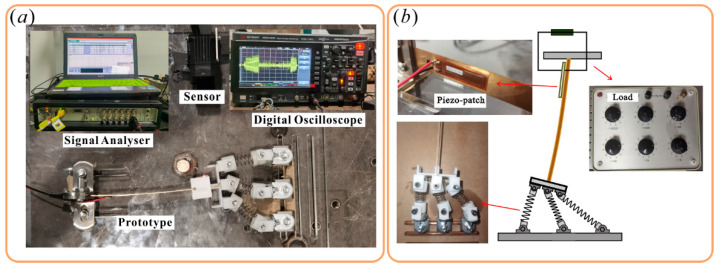
(**a**) Experimental setup; and (**b**) Prototype.

**Figure 11 sensors-22-08453-f011:**
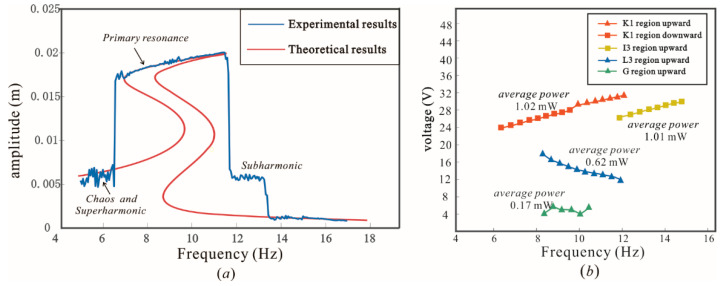
(**a**) The amplitude–response of the experimental prototype vs. theoretical results; and (**b**) Output power.

## Data Availability

The data that support the findings of this study are available from the corresponding author upon reasonable request.

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
