# Peer review of "Dynamic Design of a Quad-Stable Piezoelectric Energy Harvester via Bifurcation Theory"

_sensors, 2022, doi:10.3390/s22218453_

Round 1
Reviewer 1 Report
Please see in attachment

Author Response
Dear referee
Thanks for suggestions! We have revised the manuscript according to your comments. Our point-by-point replies are in the attachment.

Reviewer 2 Report
Dear Authors
I found your article titled: “Dynamic Design of a Quad-stable Piezoelectric Energy Harvester via Bifurcation Theory” very interesting, but in my opinion below remarks would improve your manuscript under the scientific level.
Comments and Suggestions for Authors:
1. In the Abstract please specify novelties coming from the conducted research.
2. In keywords you mentioned the geometric nonlinearity, how do you find it in your research?
3. Line 57, what a quin-stable system?
4. Such design of Energy Harvesters with piezo-beam are also used in air-flow tunnel applications. I recommend to add this information and put the suitable proposed references:
· High-performance piezoelectric wind energy harvester with Y-shaped attachments. Energy Conversion and Management, 181, pp. 645-652, 2019,
· Ceramic-based piezoelectric material for energy harvesting using hybrid excitation, Materials, 14(19), 5816, 2021.
5. Line 89, I think that word “extandable” should be substituted with “stretching”.
6. What kind of perturbation method is applied for getting the expansion of Taylor series in Appendix A?
7. What method do you use for the identification of the type of bifurcations?
8. Please specify the legend for time-series presented in Figure 4. Please mark the equilibrium points.
9. What is CDF method, please explain in the text and extend the abbreviation?
10. The experimental part is quite good presented, but I miss the information of the piezo-element producer and what kind of piezo-patch you study in the experiment?
11. Line 329, figure??
12. In Conclusions, I miss the further steps of your research, that the system has a potential to continue the research.
13. In Figure 1, there is a misspelling, and please revise the text again under the grammar and language issues.
Author Response

(The authors gave the same response as above.)

Round 2
Reviewer 1 Report
The work is significantly improved and can be considered in this form. Please remove the numbering in the conclusion.
Reviewer 2 Report
Dear Authors,
you have referred to all remarks, which I've put in. I will recommend the manuscript for its publishing in its present form.
Regards
Reviewer